# Certification of non-classicality in all links of a photonic star network without assuming quantum mechanics

Ning-Ning Wang[1,2,3,9], Alejandro Pozas-Kerstjens [4,9] ✉, Chao Zhang [1,2,3] ✉, Bi-Heng Liu[1,2,3], Yun-Feng Huang [1,2,3] ✉, Chuan-Feng Li [1,2,3] ✉, Guang-Can Guo[1,2,3], Nicolas Gisin[5,6] & Armin Tavakoli [7,8] ✉

Networks composed of independent sources of entangled particles that connect distant users are a rapidly developing quantum technology and an increasingly promising test-bed for fundamental physics. Here we address the certification of their post-classical properties through demonstrations of full network nonlocality. Full network nonlocality goes beyond standard non-locality in networks by falsifying any model in which at least one source is classical, even if all the other sources are limited only by the no-signaling principle. We report on the observation of full network nonlocality in a star-shaped network featuring three independent sources of photonic qubits and joint three-qubit entanglement-swapping measurements. Our results demonstrate that experimental observation of full network nonlocality beyond the bilocal scenario is possible with current technology.

Quantum technologies promise interesting new approaches to areas such as computing, communication, sensing and high-precision measurements. A branch that is becoming increasingly interesting is that of quantum networks. Quantum networks are infrastructures that connect distant quantum devices to each other via a given network architecture. Connections can consist in quantum communication channels or the distribution of entangled particles between different devices. The technological assets for quantum networks have been developing rapidly in recent years and many implementations of communication-oriented networks, often geared towards quantum key distribution, have been reported, see e.g., refs. [1–9]. Potential future applications of quantum networks include the idea of a quantum internet[10,11] connecting quantum devices[12], which is closely linked with the development of quantum repeaters for long-distance communication[13,14].

However, quantum networks are not only of technological interest. In the last decade, networks based on the distribution of entangled particles from multiple independent sources have become a relevant platform for studying fundamental physics (see the review article[15]). A research programme has been focused on investigating counterparts to local hidden variable models and the violation of inequalities in the spirit of Bell's theorem tailored for networks; see e.g., refs. [16–20]. The introduction of multiple independent sources in networks is known to make these scenarios considerably different from the traditional Bell experiments, which include only a single source, thereby enabling for example nonlocality[21] and device-independent randomness[22] without inputs, new forms of entanglement swapping[23], and foundational insights to quantum theory[24]. This has led to considerable interest in experimental implementations of network nonlocality[25–36]. These experiments are typically based on optical platforms, and those that

---

[1]CAS Key Laboratory of Quantum Information, University of Science and Technology of China, 230026 Hefei, China. [2]CAS Center For Excellence in Quantum Information and Quantum Physics, University of Science and Technology of China, 230026 Hefei, China. [3]Hefei National Laboratory, University of Science and Technology of China, 230088 Hefei, China. [4]Institute for Mathematical Sciences—ICMAT (CSIC-UAM-UC3M-UCM), 28049 Madrid, Spain. [5]Group of Applied Physics, University of Geneva, 1211 Geneva 4, Switzerland. [6]Constructor University, Geneva, Switzerland. [7]Physics Department, Lund University, Box 118, 22100 Lund, Sweden. [8]Institute for Quantum Optics and Quantum Information—IQOQI Vienna Austrian Academy of Sciences, Boltzmanngasse 3, 1090 Vienna, Austria. [9]These authors contributed equally: Ning-Ning Wang, Alejandro Pozas-Kerstjens. ✉e-mail: physics@alexpozas.com; drzhang.chao@ustc.edu.cn; hyf@ustc.edu.cn; cfli@ustc.edu.cn; armin.tavakoli@teorfys.lu.se

involve entanglement swapping—which is believed to be at the heart of what sets network nonlocality apart from standard Bell nonlocality[15]— are focused on the simplest network in which one party independently shares an entangled pair with each of two other parties, often also known as the bilocal network.

The violation of a network Bell inequality certifies that the network cannot be modelled exclusively by classical means. However, such a certification does not reveal much about the non-classicality of the network. For example, it is sufficient for one pair of parties, located somewhere within a large network, to perform a standard Bell test and report a violation in order for the entire network to be certified as network nonlocal[37]. Thus, it reveals only that nonlocality is present somewhere within the network. What is therefore of natural interest is to consider stronger tests of classicality, that ask whether nonlocality is present everywhere in the network, thereby certifying the non-classicality of the entire network architecture. The most basic way to do so is to separately test a number of standard Bell inequalities, one for each source involving all parties connected by it. However, this approach ignores that the sources are all independent and part of a network, and therefore also involves no entangled measurements. Recently, full network nonlocality has been put forward as a more general concept that formalises the idea of certifying nonlocality in all sources of a network[38]. Correlations in a network are called fully nonlocal if they are impossible to reproduce when at least one source in the network distributes classical physical systems. Importantly, in full network nonlocality no assumption is required that the network obeys quantum mechanics. Indeed, if correlations can be modelled by one source being classical and all other sources in the network only being constrained by the principles of no-signalling and independence (NSI), then full network nonlocality is not achieved. An interesting fact is that the largest reported quantum violation of the most widely known network Bell inequality, namely that introduced in ref. [17] for the bilocal network, is known to admit a simulation using only one nonlocal source, and thus it cannot reveal full network nonlocality[38]. Therefore, the certification of non-classicality in the entire network requires different theoretical criteria and different experimental tests. Hitherto, two demonstrations of full network nonlocality have been reported; both focused on the simplest quantum network and using the protocols proposed in the original work[38]. The full nonlocality reported in ref. [39] uses sophisticated entangled measurements but ought to be understood as a proof-of-principle since qubits that should be macroscopically separated, as in real networks, are encoded onto the same photon. The very recent demonstration reported in ref. [40] uses a simpler protocol developed in the original work[38].

However, demonstrating full network nonlocality beyond the simplest network, in a manner that both achieves the certification in an interesting way, i.e., using entangled measurements, and remains compatible with state-of-the-art photonics experiments, constitutes a challenge. Nevertheless, it is also a relevant path to embark on since the long-term aims of quantum network technology involve many sources and many parties. While there are many possible network configurations, a particularly interesting architecture for showcasing more advanced capabilities is a star configuration; a central node party is pairwise connected to $n$ separate parties via independent sources of bipartite entanglement. Star networks are a natural architecture when multiple distant users are jointly connected via a central server, and they have consequently been the focus of considerable previous theoretical work in network nonlocality[18,41–44]. While the original theoretical work[38] does propose tests of full network nonlocality in an $n = 3$ star network, the associated quantum protocol requires measurements that are exceptionally demanding on separate optical carriers and have to the best of our knowledge never been realised in any quantum mechanics context.

Here, we propose a new, tailor-made, test of full nonlocality for $n = 3$-branch star network, as illustrated in Fig. 1, and

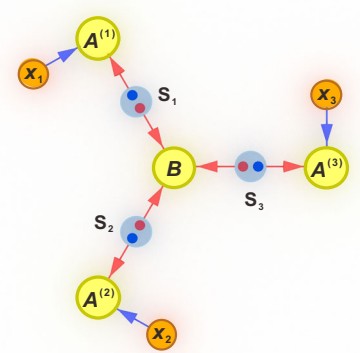

**Fig. 1 | The three-branch star network.** Bipartite physical systems, generated at the sources $S_i$, are distributed to the yellow parties, which perform measurements in the systems received. The branch parties have a choice of measurement to perform in the systems received, illustrated with the circles denoted by $x_i$.

experimentally demonstrate its relevance for optical systems in a six-photon experiment based on polarisation qubits. Our protocol leverages three-qubit entanglement swapping, but does so in a manner that is compatible with passive linear optics and achievable with a small number of optical interferences. To construct the test, we merge techniques for bounding classical[45] and NSI[46] correlations in networks. Our results serve as an early step towards the central goal of taking quantum networks that certify genuine notions of non-classicality to larger scales.

## Results
### Full nonlocality test for the star network
Consider the three-branch star network in Fig. 1. The central node party, $B$, performs a fixed measurement and outputs a result $b$. The three-branch parties, $A^{(1)}, A^{(2)}$ and $A^{(3)}$, take inputs $x_1$, $x_2$ and $x_3$, respectively, and return respective outputs $a_1$, $a_2$, and $a_3$. The network has three sources, each of which connects one of the branch parties to the central node party. It is said that the correlations $p(a_1, a_2, a_3, b | x_1, x_2, x_3)$ are network nonlocal if they cannot admit a local model in which the output of each party is influenced only by their input and a local variable, $\lambda_i$, associated with the source it connects to, namely

$$p(a_1, a_2, a_3, b | x_1, x_2, x_3)$$
$$= \sum_{\lambda_1, \lambda_2, \lambda_3} p(\lambda_1)p(\lambda_2)p(\lambda_3)p(b|\lambda_1, \lambda_2, \lambda_3)p(a_1|x_1, \lambda_1)p(a_2|x_2, \lambda_2)p(a_3|x_3, \lambda_3)$$

(1)

Full network nonlocality is a stronger notion of non-classicality. In order for the network correlations to be fully nonlocal, they must elude any model in which just one source is associated with a local variable while all the other sources are viewed as NSI. The latters are thus only constrained by the network architecture and special relativity. Notably, such no-signalling correlations are well-known to go beyond the predictions allowed in quantum theory. For simplicity, say that the source connecting $A^{(1)}$ and $B$ distributes classical physical systems, associated with a local variable $\lambda$, whereas the other two sources are NSI. The achievable correlations take the form

$$p(a_1, a_2, a_3, b | x_1, x_2, x_3) = \sum_{\lambda} p(\lambda)p(a_1|x_1, \lambda)p(a_2, a_3, b|x_2, x_3, \lambda), \quad (2)$$

where $p(\lambda)$ is the distribution of the local variable, $p(a_1|x_1, \lambda)$ is the function according to which $A^{(1)}$ produces her outcome $a_1$ given input $x_1$ and the physical system in state $\lambda$, and $p(a_2, a_3, b|x_2, x_3, \lambda)$ is a tripartite distribution only constrained by NSI. The latter implies that (i)

the choice of input of one party does not influence the output of any other party, which means that $\sum_{a_2} p(a_2, a_3, b|x_2, x_3, \lambda) = p(a_3, b|x_3, \lambda)$ and $\sum_{a_3} p(a_2, a_3, b|x_2, x_3, \lambda) = p(a_2, b|x_2, \lambda)$, and (ii) that once we marginalise the node corresponding to the central party, the joint probabilities of the unconnected parties $A^{(2)}$ and $A^{(3)}$ are independent, i.e., that $p(a_2, a_3|x_2, x_3) = \sum_{b,\lambda} p(\lambda) p(a_2, a_3, b|x_2, x_3, \lambda) = p(a_2|x_2) p(a_3|x_3)$. In order to certify full network nonlocality, we similarly insist that no model of the form Eq. (2) is possible also when we permute the position of the classical source to any one of the three possible configurations.

Characterising the distributions that take the form of Eq. (2) is hard due to the well-known non-convexity of correlations in networks with independent sources. However, techniques have been developed for relaxing problems in this spirit; providing sequences of necessary conditions for a probability distribution admitting a classical[45], quantum[47] or NSI[46] model in a given network. Our purpose, namely to falsify a model of the form Eq. (2) (up to permutations of the position of the classical source) can be viewed as a hybrid situation between the classical and NSI network correlation problems. We can combine these so-called inflation tools to analyse our problem[38]. The heart of the idea of these methods is to consider a larger network than the star configuration, which is built from copies of the components (namely, $p(\lambda)$, $p(a_1|x_1, \lambda)$, and $p(a_2, a_3, b|x_2, x_3, \lambda)$) of the network of our interest. It is then possible to formulate a necessary condition for a model Eq. (2) as a linear program, which can be solved using standard methods. The larger the inflation of the original network, the more demanding becomes the linear program and the more accurate becomes the relaxation. If a candidate distribution fails to pass the linear programming test for a given inflation, it cannot admit a model Eq. (2). If the same distribution also fails to pass the analogous inflation test for the other two possible configurations of the classical source, then it is certified as fully network nonlocal. Importantly, for each of the three cases, one can obtain a certificate of the failure of the hypothesised model by evaluating the dual of the corresponding linear program[48]. This comes in the form of an inequality in the probability elements $p(a_1, a_2, a_3, b|x_1, x_2, x_3)$, which is violated by the candidate distribution. Such a criterion can also detect full network nonlocality for other distributions and thus also applies to noisy circumstances.

In this work, we consider the distribution $p(a_1, a_2, a_3, b|x_1, x_2, x_3)$ that is generated in Fig. 1 when the three sources of the network distribute maximally entangled qubit states, $|\phi^+\rangle = (|00\rangle + |11\rangle)/\sqrt{2}$, the central party performs a two-outcome entanglement-swapping measurement on their three qubits given by $\{|GHZ\rangle\langle GHZ|, \mathbb{1} - |GHZ\rangle\langle GHZ|\}$ with $|GHZ\rangle = (|000\rangle + |111\rangle)/\sqrt{2}$ being the Greenberger-Horne-Zeilinger (GHZ) state, and the branch parties each choose one of the same two measurements, namely

$$A_0^{(1)} = A_0^{(2)} = A_0^{(3)} = \sin\theta_0 \sigma_X + \cos\theta_0 \sigma_Z,$$
$$A_1^{(1)} = A_1^{(2)} = A_1^{(3)} = \sin\theta_1 \sigma_X + \cos\theta_1 \sigma_Z,$$
(3)

where $\sigma_X$ and $\sigma_Z$ denote the corresponding Pauli matrices. Following the hybrid inflation method, for some values of $\theta_0$ and $\theta_1$, the resulting distribution becomes fully network nonlocal. We have systematically considered different choices of the angles $(\theta_0, \theta_1)$ and found that the best choice of angles is $\theta_0 \approx -1.865$ and $\theta_1 \approx -0.4146$. In a moment we will see how they emerge from our explicit full network nonlocality test. The proof of full network nonlocality is given by the impossibility of finding a suitable distribution in a concrete inflation of the three-branch star network, which is described in the "Methods" section. Analysing the certificate of infeasibility in the linear programme associated with the inflation, based on the above angles, one can

extract the following criterion satisfied by the model Eq. (2),

$$
\begin{aligned}
\mathcal{I}_1 = &-\langle A_0^{(1)} A_0^{(2)} A_0^{(3)} B\rangle - \langle A_1^{(1)} A_0^{(2)} A_0^{(3)} B\rangle - \langle A_0^{(1)} A_0^{(2)} A_1^{(3)} B\rangle + \langle A_1^{(1)} A_0^{(2)} A_1^{(3)} B\rangle \\
&-\langle A_0^{(1)} A_0^{(2)} A_0^{(3)}\rangle - \langle A_1^{(1)} A_0^{(2)} A_0^{(3)}\rangle - \langle A_0^{(1)} A_0^{(2)} A_1^{(3)}\rangle + \langle A_1^{(1)} A_0^{(2)} A_1^{(3)}\rangle - \langle A_0^{(1)} A_0^{(2)} B\rangle \\
&-\langle A_1^{(1)} A_0^{(3)} B\rangle - \langle A_0^{(1)} A_1^{(3)} B\rangle + \langle A_1^{(1)} A_1^{(3)} B\rangle - \langle A_0^{(1)} A_0^{(3)}\rangle - \langle A_1^{(1)} A_0^{(3)}\rangle - \langle A_0^{(1)} A_1^{(3)}\rangle \\
&+\langle A_1^{(1)} A_1^{(3)}\rangle - 2\langle A_0^{(2)} B\rangle - 2\langle A_0^{(2)}\rangle - 2\langle B\rangle - 2 \\
\leq &\ 0,
\end{aligned}
$$
(4)

where $\langle A_{x_1}^{(1)} A_{x_2}^{(2)} A_{x_3}^{(3)} B\rangle = \sum_{a_1, a_2, a_3, b} (-1)^{a_1 + a_2 + a_3 + b} p(a_1, a_2, a_3, b|x_1, x_2, x_3)$, and analogously for the remaining correlators by removing the corresponding factors inside the sum. Thus, any distribution $p(a_1, a_2, a_3, b|x_1, x_2, x_3)$ that violates Eq. (4) cannot be realised by having a classical source connecting $A^{(1)}$ and $B$ and two NSI sources between $B$, and $A^{(2)}$ and $A^{(3)}$, respectively.

The inequality Eq. (4) can be interpreted as the CHSH inequality between parties $A^{(1)}$ and $A^{(3)}$ when both remaining parties output 0 and party $A^{(2)}$ measures $x_2 = 0$. Thus, it is fundamentally different from tests of non-classicality of a single source by violating a Bell inequality between one branch party and the central one. Moreover, in order to violate Eq. (4), not only (at least) several sources must be non-classical, but also the central party must perform an entangling measurement.

The symmetry of the network and the candidate probability distribution allow us to obtain inequalities analogous to Eq. (4) also for the two remaining arrangements of the classical source. Similarly, the violation of these inequalities witness the non-classicality of the sources connecting $B$ with $A^{(2)}$ and $A^{(3)}$, respectively, by performing the cyclic permutations $A^{(1)} \to A^{(2)} \to A^{(3)} \to A^{(1)}$ and $A^{(1)} \to A^{(3)} \to A^{(2)} \to A^{(1)}$, giving rise to inequalities $\mathcal{I}_2 \leq 0$ and $\mathcal{I}_3 \leq 0$. Therefore, a simultaneous violation of all three inequalities $\mathcal{I}_i \leq 0$ for $i = 1, 2, 3$ implies that no source in the network admits a classical description, i.e., full network nonlocality is observed.

In the quantum model, due to permutation symmetry for the parties in our chosen strategy, all three values of $\mathcal{I}_i$ will be identical. We can evaluate this value for any pair of angles $(\theta_0, \theta_1)$. Moreover, for the purposes of the experiment, we also consider a simple noise model in which the sources are assumed to emit isotropic states $v|\phi^+\rangle\langle\phi^+| + (1-v)\mathbb{1}/4$ with visibility $v$. The quantum value of the Bell parameter then becomes

$$\frac{v^2}{4}\left(v\sin\theta_0\left(\sin^2\theta_1 - 2\sin\theta_0\sin\theta_1 - \sin^2\theta_0\right) - 2\cos\theta_0\cos\theta_1 - \cos^2\theta_0 + \cos^2\theta_1 - \frac{2}{v^2}\right)$$
(5)

The maximal value is 0.1859 and occurs at $\theta_0 \approx -1.865$ and $\theta_1 = -0.415$. For this choice, violation is achieved whenever $v \approx 0.882$. This noise tolerance is crucial for experimental purposes. Notably, the visibility can be marginally lowered to $v \approx 0.881$ by choosing angles $\theta_0 = -1.908$ and $\theta_1 \approx -0.367$, which for a noiseless setting give the slightly suboptimal violation 0.1843. In a similar way, by considering more general measurements, $A_j^{(i)} = \sin\theta_j \cos\phi_j \sigma_X + \sin\theta_j \sin\phi_j \sigma_Y + \cos\theta_j \sigma_Z$, one can achieve the larger violation of the inequalities of $(\sqrt{2} - 1)/2 \approx 0.2071$ (for $\theta_0 = \theta_1 = \pi/2$ and $\phi_0 = \phi_1/3 = \pi/4$), at the cost of a larger critical visibility of $v = 2^{-1/6} \approx 0.891$.

## Optical three-branch quantum star

We build the quantum star network by using three "sandwich-like" spontaneous parametric down-conversion (SPDC) sources. Each source produces polarisation-entangled photon pairs to supply the entangled photonic link. The experimental setup is illustrated in Fig. 2. Three photons (each from a source) are sent to the central node $B$, while the other three photons are distributed to three-branch parties $A^{(1)}$, $A^{(2)}$, and $A^{(3)}$, respectively, to construct the star network. The qubits are encoded in the polarisation degree of freedom of the photons. Our

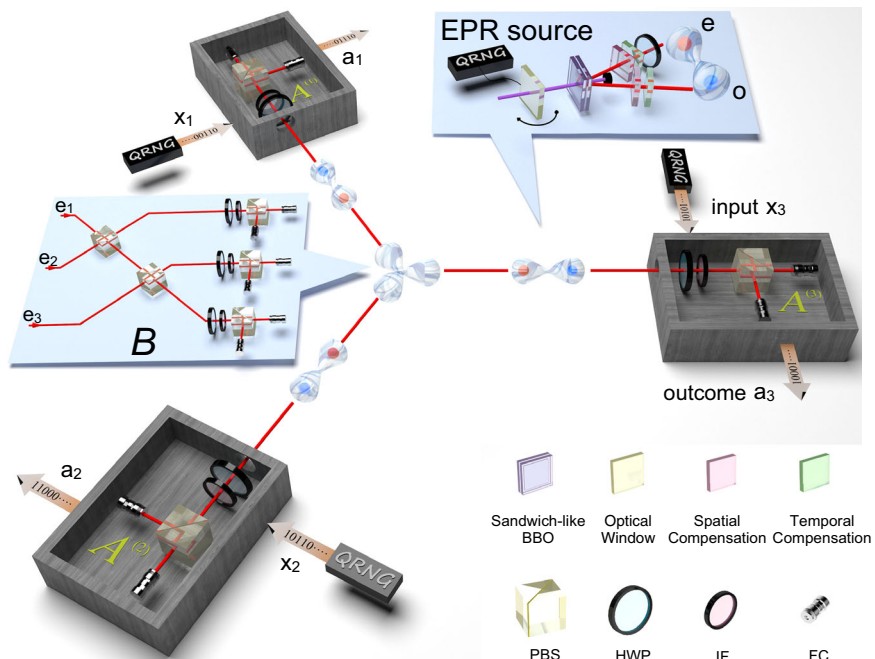

**Fig. 2 | Sketch of the experimental setup.** Three SPDC sources distribute maximally entangled photon pairs to build a 3-star quantum network. The three extraordinary photons (red spheres) sent to the central node are spatially and temporally overlapped on two cascaded PBSs to complete the Hong-Ou-Mandel interference, as shown in the left insert. The three ordinary photons (blue spheres) are sent to the wing parties to realise single-qubit measurements in Eq. (3). The SPDC source is based on a sandwich-like BBO-HWP-BBO crystal and is pumped by an ultraviolet pump pulse (390-nm, 80-MHz, 140-fs). A randomly rotated glass slice is inserted before each source to randomise the phase of the laser pulse before hitting on the sandwich crystal. PBS polarisation beam splitter, HWP half-wave plate, IF interference filter, FC fibre coupler, QRNG quantum random number generator, BBO beta barium borate.

sandwich-like Einstein-Podolsky-Rosen (EPR) sources employ beam-like type-II phase matching, which achieves high brightness (0.3 MHz), high fidelity (98%) and high collection efficiency (40%) at the same time. The high performance of the photon sources is crucial to the success of the experiment.

The prerequisite for certifying nonlocality in a quantum network is that all the photon sources in the network are independent. To meet this requirement in the experiment, we split a single laser beam into three and pump three SPDC sources in parallel. Photon pairs are thus generated in separate nonlinear crystals. However, as the pump beams are generated by the same laser, correlations may still be present. In order to improve the independence, we insert a randomly rotated glass slice (controlled by independent quantum random number generators) before each source to randomise the phase of the pump beam. We set the angle of the glass slice to refresh every ~20 ms, which is much faster than the six-fold coincidence rate (~0.5 Hz) in our experiment, thus effectively erasing any coherence between the three pump beams on the time scale of the network.

At the central node, we extend the widely used Bell-state measurement device to three qubits, as shown in the insert of Fig. 2. The received photons are injected into an optical interferometer, which consists of two cascaded PBSs and three 22.5° HWPs at each output. Delay lines and interference filters are introduced to make the interfering photons indistinguishable in arrival time and spectrum. If we postselect the output case when there is one and only one photon in each output (with a success probability of 1/4), the input state can be projected into $|GHZ\rangle$ or $|GHZ^-\rangle = (|000\rangle - |111\rangle)/\sqrt{2}$ according to different detected events. When the GHZ-state projection is successful, we record the events when all wing parties detect one photon and then obtain a six-fold coincidence. The measurement device at each branch party is a polarisation analysis system, which consists of a HWP, a PBS and two fibre-coupled single-photon detectors. The HWP is mounted on a rotation stage and controlled by a QRNG. When the input is 0 (1), the HWP will be set at $\theta_0/4$ ($\theta_1/4$), which can project

the input state into the eigenbasis of the $A_0$ ($A_1$) measurement operator, and the output 0 (1) is recorded when the transmitted (reflected) detector fires.

## Experimental results

During the experiment, we switch the measurement settings of the three wing parties every 15 s (excluding the rotation time of the motors), and collect a total of 155019 six-photon coincidence events in 19200 switching cycles. The measured results of $p(a_1, a_2, a_3|b = 0, x_1, x_2, x_3)$ are shown in Fig. 3. Then we measure the projection probability $p(b = 0)$. The measured result is $0.1297 \pm 0.0027$ where the standard deviation represents statistical error. The value slightly larger than the theoretical prediction, $1/8 = 0.125$, is due to the systematic error of higher-order emission noise in our system. Finally we calculate $p(a_1, a_2, a_3, b = 0|x_1, x_2, x_3) = p(a_1, a_2, a_3|b = 0, x_1, x_2, x_3) \times p(b = 0)$ and the values for the inequalities $\mathcal{I}_i$, which correspond to $\mathcal{I}_1 = 0.0598 \pm 0.0041$, $\mathcal{I}_2 = 0.0404 \pm 0.0040$ and $\mathcal{I}_3 = 0.0471 \pm 0.0041$. The values obtained exceed the corresponding non-FNN bounds by more than 10 standard deviations. The $p$-values associated with the violations of the three inequalities are $1.143 \times 10^{-49}$, $1.747 \times 10^{-25}$, and $4.864 \times 10^{-32}$, respectively. Note that our experimental demonstration is subject to the common loopholes in nonlocality experiments, namely the locality loophole and the postselection loophole[49].

Via state tomography, we find that the fidelity of the three EPR sources is $0.9793 \pm 0.0001$, $0.9788 \pm 0.0001$ and $0.9811 \pm 0.0001$, respectively. Similarly, we find via measurement tomography a fidelity of $0.8205 \pm 0.0040$ for the three-qubit GHZ projection in the node. Finally, to evidence the independence of the sources not only in the experimental setup but also in the data, we have performed additional measurements, experimentally determining the marginal probabilities $p(a_i, a_j|x_i, x_j)$, and computed their mutual information[31,39]. As we show in Supplementary Table 1, we find that the mutual information very nearly vanishes in all cases, as expected from truly independent sources.

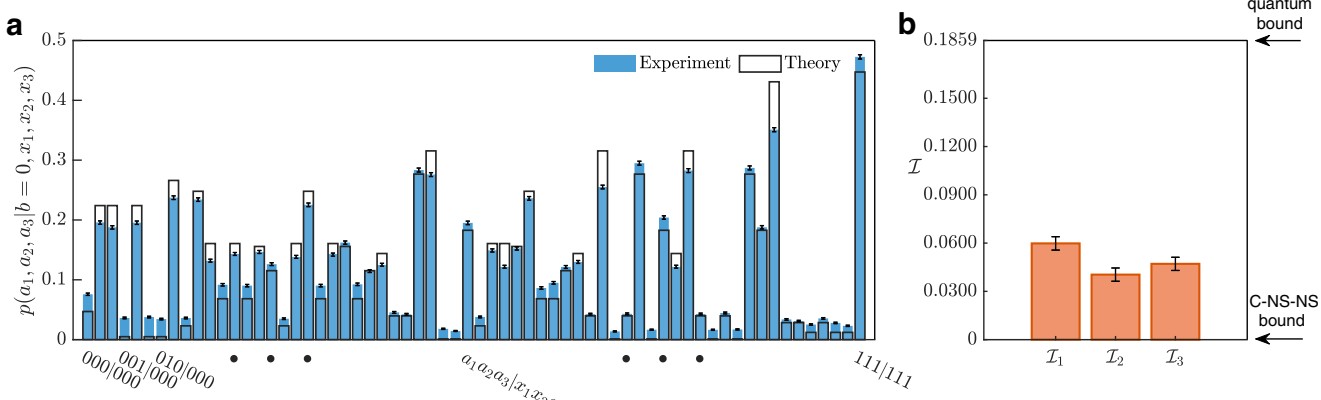

**Fig. 3 | Experimental results. a** Estimation of the branch parties' probability distributions conditioned to a successful GHZ-state projection in the central node. The distributions are obtained by normalising the raw experimental data in each measurement setting. **b** The measured results for the three inequalities for testing full network nonlocality. The theoretical maximum value is equal to 0.1859 (the quantum bound) for all inequalities. The error bars represent one standard deviation and are deduced from the photon statistical error of the raw data.

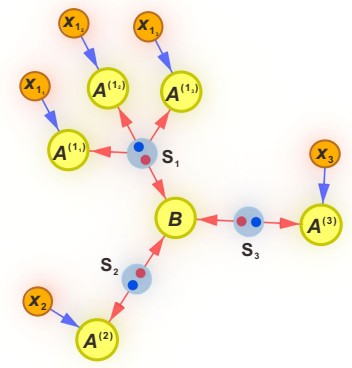

**Fig. 4 | The inflation of the three-branch star network used for obtaining Eq. (4).** Assuming that the source connecting to $A^{(1)}$ is classical, one can clone the information sent to it and distribute the copies to copies $A^{(1)}$.

## Discussion

Guaranteeing non-classical properties in quantum networks is an increasingly relevant problem and, as is also the case with more elementary bipartite systems, nonlocality provides a powerful avenue for this purpose. Whereas standard network nonlocality is known to be insufficient for guaranteeing the non-classical properties of a whole network architecture, the stronger concept of full network nonlocality better meets the challenge. Here, we have reported on an optical demonstration of full network nonlocality in a star-shaped network involving four parties, three independent sources of EPR pairs and three-qubit entanglement-swapping measurements. The realisation of strong forms of quantum correlations in more complex networks is also a motivation for the development of quantum information protocols for parties limited by a network architecture. This direction of research, with notable exceptions[47,50], is mostly unexplored. Importantly, our demonstration showcases full network nonlocality without assuming that the sources in the network are limited by quantum mechanics, thus achieving the certification of non-classicality without requiring knowledge of a physical model. We note that an alternative avenue also is possible in which the nonlocal resources are assumed to be limited by quantum mechanics. This will lead to a less fundamentally motivated certification, but still suitable for quantum networks, having less demanding fidelity requirements in experiments.

Our results constitute a foundational step in taking more genuine forms of optical network nonlocality beyond the simplest four-photon scenario. On the theoretical side, we have showed not only the ability of hybrid inflation methods to provide nontrivial tests for larger networks, but also that these can be tailored for experimentally friendly protocols. On the experimental side, we have shown that a six-photon network with multi-qubit entanglement swapping can be implemented at a sufficiently high quality and efficiency to pass these comparatively demanding tests of classicality. Nevertheless, our experiment leaves open the detection and locality loopholes, but recent experiments based on the simplest network[28,51,52] offer a potential avenue towards closing the latter loophole also for larger networks by carefully synchronising the signals from the independent sources. Similarly, more stringent implementations of independent sources, that do not rely on the quantum-inspired idea of a randomised phase, have been implemented separately[34,53,54] and their incorporation into the larger network considered here and beyond is a natural next step.

In addition to the above, two more basic challenges for proceeding to nonlocality and entanglement-swapping tests in networks larger than ours are (i) the production of multipartite quantum states at a viable rate and (ii) the implementation of high-quality multi-qubit entanglement-swapping at a viable rate. Whereas sophisticated forms of multi-photon entanglement have been reported[55], the associated rates are not well-suited for network considerations. Notably, however, recent advances based on time-multiplexing and feed-forward[56] offer a path to better rates. Moreover, deterministic entanglement swapping with passive linear optics is many times not possible without auxiliary photons[57] and, while probabilistic entanglement swapping, as in our experiment, is possible, it typically comes with success rates rapidly decreasing in the number of qubits[58]. An potentially interesting path towards larger quantum networks is then to consider light-matter entanglement and more standard deterministic multi-qubit entanglement swapping[31].

## Methods

### Full network nonlocality inequalities from inflation

Inflation allows to derive necessary conditions for distributions compatible with a given network by analysing hypothetical scenarios where one had access to multiple copies of the elements in the network. For instance, take the network in Fig. 1, and assume that $S_I$ is a source of classical shared randomness. Since classical shared randomness can be cloned, one could make multiple copies of it and send them to multiple

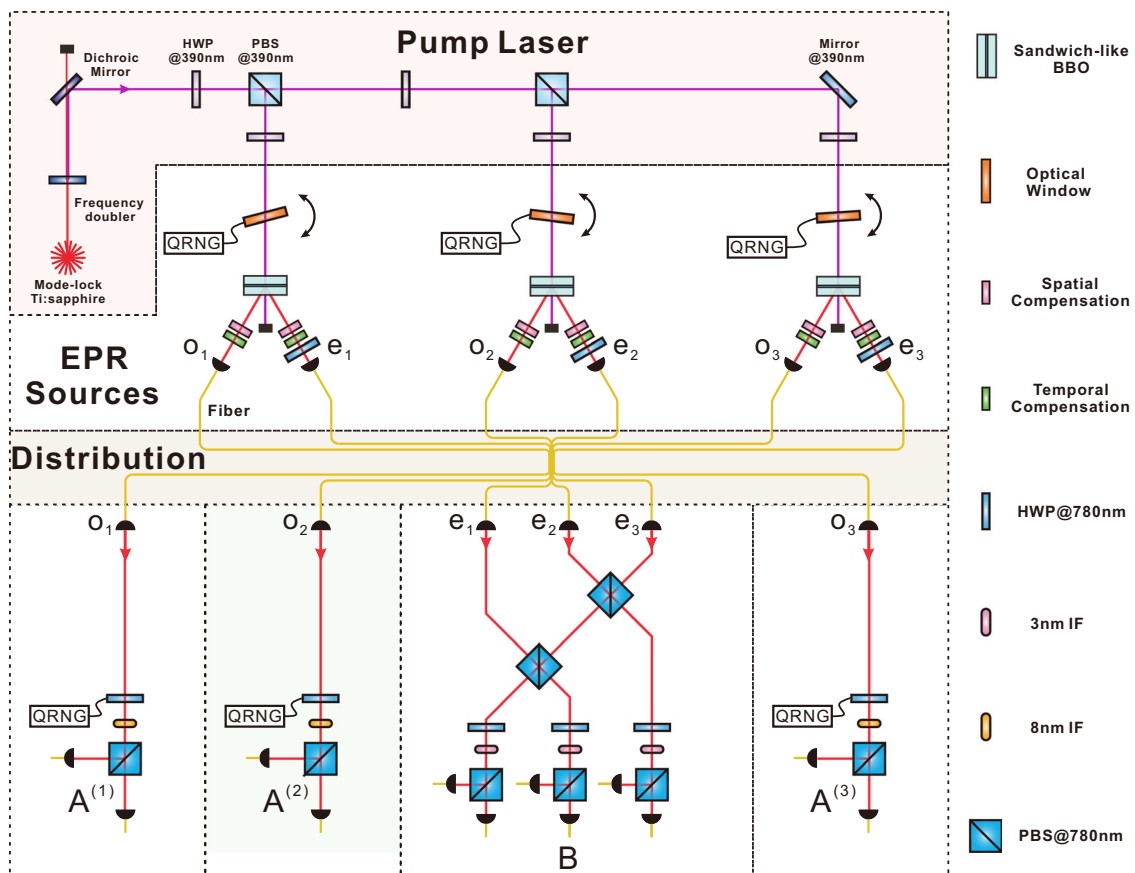

**Fig. 5 | Detailed experimental setup.** An ultraviolet pulse is generated by a frequency doubler and is divided into three parallel beams. The relative phases between these three beams are erased before they hit the sandwich crystals. Ordinary photons from the EPR sources are sent to the branch parties and different measurement settings are selected according to the random input. Extraordinary photons are projected into the GHZ state at the central node. PBS polarisation beam splitter, HWP half-wave plate, IF interference filter, QRNG quantum random number generator, BBO, beta barium borate.

copies of the party $A^{(1)}$ as in Fig. 4. Any distribution generated in this new network, $p_{inf}(a_{1_1}, a_{1_2}, a_{1_3}, a_2, a_3, b|x_{1_1}, x_{1_2}, x_{1_3}, x_2, x_3)$, would satisfy:

(i) Positivity and normalisation, namely

$$p_{inf}(a_{1_1}, a_{1_2}, a_{1_3}, a_2, a_3, b|x_{1_1}, x_{1_2}, x_{1_3}, x_2, x_3) \geq 0 \qquad (6)$$

$$\sum_{\substack{a_{1_1}, a_{1_2}, a_{1_3} \\ a_2, a_3, b}} p_{inf}(a_{1_1}, a_{1_2}, a_{1_3}, a_2, a_3, b|x_{1_1}, x_{1_2}, x_{1_3}, x_2, x_3) = 1 \qquad (7)$$

(ii) No-signalling between the parties, i.e.,

$$\sum_{a_{1_1}} p_{inf}(a_{1_1}, a_{1_2}, a_{1_3}, a_2, a_3, b|x_{1_1}, x_{1_2}, x_{1_3}, x_2, x_3)$$
$$- \sum_{a_{1_1}} p_{inf}(a_{1_1}, a_{1_2}, a_{1_3}, a_2, a_3, b|\tilde{x}_{1_1}, x_{1_2}, x_{1_3}, x_2, x_3) = 0, \qquad (8)$$

for any values of $x_{1_1}$ and $\tilde{x}_{1_1}$, and analogously for the rest of the parties.

(iii) Since $A^{(1_1)}, A^{(1_2)}$ and $A^{(1_3)}$ are all copies of the same party, $A^{(1)}$, and in every round they all receive the same information from the local hidden variable, they all produce the same outcome in every round, and thus the distribution $p_{inf}$ is invariant under relabellings of $A^{(1_1)}, A^{(1_2)}$ and $A^{(1_3)}$, namely

$$p_{inf}(a_{1_1}, a_{1_2}, a_{1_3}, a_2, a_3, b|x_{1_1}, x_{1_2}, x_{1_3}, x_2, x_3)$$
$$- p_{inf}(a_{1_{\pi(1)}}, a_{1_{\pi(2)}}, a_{1_{\pi(3)}}, a_2, a_3, b|x_{1_{\pi(1)}}, x_{1_{\pi(2)}}, x_{1_{\pi(3)}}, x_2, x_3) = 0 \qquad (9)$$

for any permutation $\pi \in \{1 \leftrightarrow 2, 1 \leftrightarrow 3, 2 \leftrightarrow 3, 1 \rightarrow 2 \rightarrow 3 \rightarrow 1, 1 \rightarrow 3 \rightarrow 2 \rightarrow 1\}$.

(iv) When marginalising two of the copies of $A^{(1)}$, the network is the original one of Fig. 1, and thus

$$\sum_{a_{1_2}, a_{1_3}} p_{inf}(a_{1_1}, a_{1_2}, a_{1_3}, a_2, a_3, b|x_{1_1}, x_{1_2}, x_{1_3}, x_2, x_3) = p(a_1, a_2, a_3, b|x_1, x_2, x_3). \qquad (10)$$

The set of conditions (6)–(10) are necessary for a distribution $p(a_1, a_2, a_3, b|x_1, x_2, x_3)$ to be compatible with Fig. 1, but they may not be sufficient. For instance, we have not imposed that the marginalisations over the central party factorise. In any case, if no $p_{inf}$ exists that satisfies equations (6)–(10), then it is not possible to generate the $p(a_1, a_2, a_3, b|x_1, x_2, x_3)$ under scrutiny in the original network of Fig. 1.

The set of equalities (6)–(10) can be written in a convenient form as $A \cdot p_{inf} \geq b$, where the entries of the vector $p_{inf}$ are the elements of $p_{inf}(a_{1_1}, a_{1_2}, a_{1_3}, a_2, a_3, b|x_{1_1}, x_{1_2}, x_{1_3}, x_2, x_3)$, the matrix $A$ contains the coefficients associated with the probabilities in the left-hand sides of Eqs. (6)–(10), and the vector $b$ contains the corresponding right-hand sides. This means that the problem of finding a $p_{inf}$ satisfying conditions (6)–(10) can be solved using linear programming. Importantly, in linear programming the impossibility of finding a feasible solution comes accompanied by a vector that satisfies $y \cdot A = 0$ and $y \cdot b > 0$[48]. Since in Eqs. (6)–(10) the coefficients of $A$ are independent of $p$ and thus $y \cdot A = 0$ always, one can understand the inequality $y \cdot b > 0$ as a witness, whose satisfaction by a particular distribution signals it as FNN. As we show in the computational appendix[59], the inequality in Eq. (4) is the witness associated with the linear programme defined by Eqs.

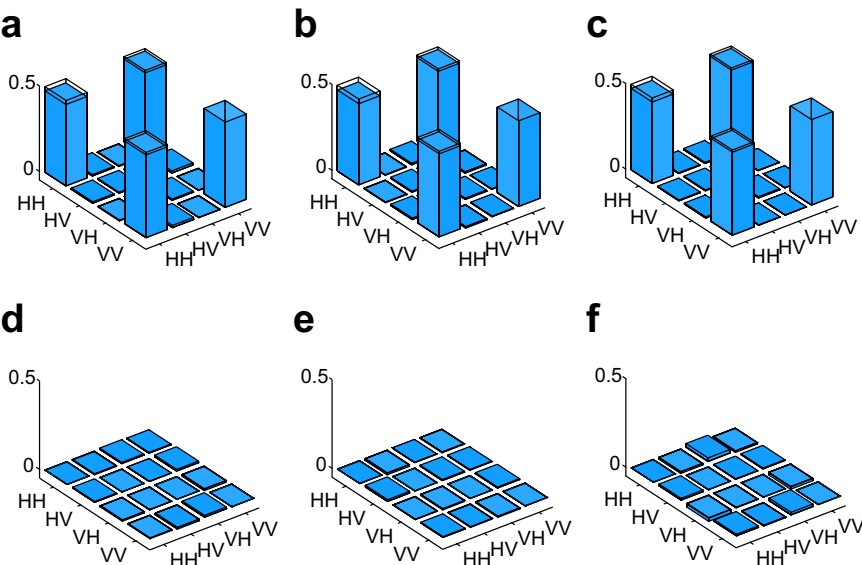

**Fig. 6 | Tomographic results of the three EPR sources.** The top and bottom plots in each column show, respectively, the real and imaginary parts of the reconstructed density matrices corresponding to the states distributed by the sources (left) $S_1$, (centre) $S_2$, and (right) $S_3$ in Fig. 1. This is, the panels correspond to **a** $S_1$, real part; **b** $S_2$, real part; **c** $S_3$, real part; **d** $S_1$, imaginary part; **e** $S_2$, imaginary part; **f** $S_3$, imaginary part.

(6)–(10), with a global change of sign so that FNN is witnessed by the violation of the inequality.

## Experimental details

Here we describe the detailed experimental implementation of the optical three-branch star network. The detailed experimental setup is shown in Fig. 5.

**EPR source.** The three sources of EPR states are generated by a same mode-locked Ti:sapphire laser. The parent laser pulse has a central wavelength of 779 nm, a duration of 140 fs, and a repetition rate of 80 MHz, that firstly passes through a frequency doubler in order to generate an ultraviolet pump pulse. This pump pulse is split into three beams of equal power by means of polarisation beam-splitters and half-wave plates. Each of the three split beams has a pump power of 260 mW, and is used to pump an independent spontaneous parametric down-conversion (SPDC) source that generates the EPR states. The SPDC source is based on a sandwich-like structure composed of a true-zero-order half-wave plate (THWP) in between of two 2mm-thick beta barium borate (BBO) crystals. The two BBO crystals are identically cut for beam-like type-II phase matching, in order to benefit from the high brightness and high collection efficiency of the source. The pump photon has equal probability to be downconverted in both BBO crystals, which both produce photon pairs in the polarisation state $|H\rangle_1|V\rangle_2$. If the pair is generated in the first crystal, the passing through the THWP rotates it to the state $|V\rangle_1|H\rangle_2$. Therefore, after spatial and temporal compensations, the produced state is $(|H\rangle_1|V\rangle_2 + |V\rangle_1|H\rangle_2)/\sqrt{2}$. Then we use a HWP to transform the state to the Bell state $(|HH\rangle + |VV\rangle)/\sqrt{2}$ required for the experiment. For doing so, we use a spectral filter of 3 nm for each extraordinary photon ($e_i$ in Fig. 5) and of 8 nm for each ordinary photon ($o_i$ in Fig. 5). The counting rate for each source is about 0.3 MHz and collection efficiency is about 40%.

To characterise the EPR sources, we perform state tomography of each source. The reconstructed matrices are shown in Fig. 6. The fidelity of the states $F = \langle\phi^+|\rho_{exp}|\phi^+\rangle$ are calculated to be $0.9793 \pm 0.0001$, $0.9788 \pm 0.0001$, $0.9811 \pm 0.0001$, respectively.

**Source independence.** The pump beams to the three EPR sources come from the same laser pulse, which could insert unintended correlations between them. In order to ensure that the three sources are independent, we insert a randomly rotated glass slice in each pump beam before hitting the SPDC sources. This destroys any coherence between the beams. The N-BK7 glass slices have a thickness of 5 mm and are calibrated for normal incidence of the pump beam. The glass slice is mounted on a motorised rotation stage and controlled by a QRNG, which generates random real numbers in the interval [0, 1]. These random numbers are mapped uniformly to rotation angles in [0°, 0.6°] with resolution of 0.01°, corresponding to random phase shifts between 0 and $2\pi$. The maximal rotation of the glass slices, namely 0.6°, introduces an optical path change of ~410 nm. We refresh the angle of each glass slice every ~20 ms, which is much faster than the six-fold coincidence rate in the experiment (~0.5 Hz), thus effectively erasing any relative phase between the pump beams on the time scale of the network. The measured degree of source independence can be found in the Supplementary Note 2.

**GHZ measurement.** The central party in the star network performs a two-outcome GHZ measurement, which consists of the projectors $\{|GHZ\rangle\langle GHZ|, \mathbb{1} - |GHZ\rangle\langle GHZ|\}$. We implement this measurement, as depicted in the bottom part of Fig. 5, using two cascaded polarising beam-splitters (PBSs) and three 22.5° HWPs, one at each output. In this device, the projection onto $|GHZ\rangle\langle GHZ|$ corresponds to having one detection at each port, with none or two of them in the reflected detectors. We characterise this measurement in Fig. 7 via measurement tomography, finding a fidelity to the ideal projector onto the GHZ state of $0.8205 \pm 0.0040$. When considering ideal sources but an imperfect GHZ measurement with white noise, the required fidelity for the GHZ measurement in order to observe a violation of Eq. (4) (and its corresponding versions for the remaining situations) is 0.763.

Note that only terms corresponding to the projection onto $|GHZ\rangle\langle GHZ|$ are necessary to compute the quantities $\mathcal{I}_i$. However, the measurement device built is also capable of detecting a projection onto $|GHZ^-\rangle\langle GHZ^-|$, which corresponds to having one detection at each port, with one of all of them in the reflected detectors. This allows to use our device for more complicated protocols where the central party performs a three-outcome measurement.

**Estimation of $p(b = 0)$.** The GHZ measurement device employed does not use photon-number resolved detectors, and thus is unable to

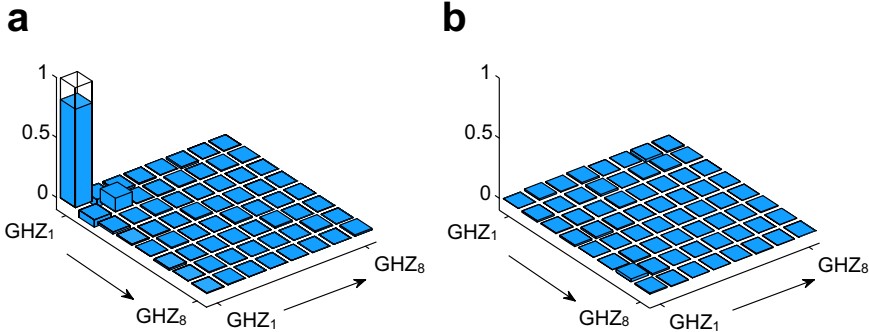

**Fig. 7 | Tomographic results of the GHZ projector.** The **a** real part and **b** imaginary part of the reconstructed matrix of the GHZ projector.

detect the events where two photons go into the same output port. This has the consequence that the configuration of the device with the HWPs at 22.5° cannot distinguish between events where the projection is performed onto another state and events where photons are lost. In order to calculate the number of successful runs (i.e., the number of events where no photons are lost) and to estimate $p(b = 0)$ by #(GHZ events)/#(succ. runs), we rotate the three HWPs in the central node to 0° and record the number of six-photon events in the whole experiment (three in the central node, and one more for each branch party). In order to reduce the experimental fluctuations, we switch between the three HWPs being at 0° or at 22.5° every 60 s. After 2 h of measurement, we collect 15562 total six-photon events and 2019 successful projection events, thus estimating the probability of projection onto the |GHZ⟩ state to be $p(b = 0) = 0.1297 \pm 0.0027$.

## Data availability

The raw data that support the findings of this study are mainly available in Supplementary Information. Additional data are available from the corresponding authors upon request.

## Code availability

The code used to construct the full network nonlocality inequalities is available at https://www.github.com/apozas/three-star-fnn.

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

## Acknowledgements

This work was partially carried out at the USTC Centre for Micro and Nanoscale Research and Fabrication. This research was supported by the Innovation Programme for Quantum Science and Technology (No. 2021ZD0301604), the National Natural Science Foundation of China (Nos. 11821404, 11734015, 62075208), the Fundamental Research Funds for the Central Universities (nos. WK2030000061, YD2030002015) (C.Z., B.-H.L., Y.-F.H., C.-F.L.), the Spanish Ministry of Science and Innovation MCIN/AEI/10.13039/ 501100011033 (CEX2019-000904-S and PID2020-113523GB-I00), the Spanish Ministry of Economic Affairs and Digital Transformation (project QUANTUM ENIA, as part of the Recovery, Transformation and Resilience Plan, funded by EU programme NextGenerationEU), Comunidad de Madrid (QUITEMAD-CM P2018/TCS-4342), the CSIC Quantum Technologies Platform PTI-001 (A.P.-K.), the Swiss National Science Foundation via the NCCR-SwissMap (N.G.), the Wenner-Gren Foundation, and the Knut and Alice Wallenberg Foundation through the Wallenberg Centre for Quantum Technology (WACQT) (A.T.).

## Author contributions

N.-N.W., C.Z. and Y.-F.H. conceived the experiments and analysed the data with discussion of B.-H.L., C.-F.L. and G.-C.G.; N.-N.W. performed the experiments with assistance of C.Z.; A.P.-K. constructed the theoretical protocol with discussion of N.G. and A.T.; A.P.-K., C.Z. and A.T. wrote the manuscript; Y.-F.H., C.-F.L., G.-C.G., N.G. and A.T. supervised the project. All authors read the paper and discussed the results.

## Competing interests

The authors declare no competing interests.
