## [Peer Review File · Nature Communications]

Certification of non-classicality in all links of a photonic star network without assuming quantum mechanicsREVIEWERS' COMMENTS

Reviewer #1 (Remarks to the Author):

Wang et al. report on the certification of non-classicality within an experimental network having a star-shaped configuration. By falsifying any model with at least one classical source, this work aims at the certification of "full network nonlocality" (FNN). The work has been experimentally implemented using simultaneously three entangled photon sources (SPDC) for the generation of the photons which encodes the information in the polarization degree of freedom. This is a challenging and interesting experiment. To definitively test nonlocality in a network all the sources should be independent and independently pumped. This would be an extremely complex configuration to realize. Accordingly, the authors exploited a single laser to pump the three crystals. Since the independence of the source was not guaranteed due to the single laser, the authors improved the assumption by using randomly rotated glasses.

The paper is clearly written, the experimental procedure seems sound, the error analysis rigorous, and the conclusions appear justified. This work certainly deserves publication in a broad-impact journal. The results, all-in-all, in my opinion represent a sufficient advance with respect to the state of the art that can justify publication in Nature Communications. As stated by the authors, full network non-locality has been already experimentally demonstrated only in bilocal systems (arXiv:2201.06361). Despite not using truly independent sources this article still represents a relevant contributions within the general scenario of quantum information.

Therefore, I can recommend publication of the manuscript in Nature Communications once the following points are clearly addressed.

- 1) The authors should comment on the postselection performed when the Bell state measurement is carried out. In particular about the device-dependent nature of the experiment.
- 2) It would be useful for the reader to have in Fig. 3b the theoretical values expected for the inequalities exploited.
- 3) It is not clear to me which fidelity is required to have a simultaneous violation of the inequalities. In particular, on page 10 it is discussed a visibility $v=0.88$ and $v=0.89$ while the tomography reveals a visibility of the GHZ state of $v=0.82$. What is the connection between the experimental visibility and the one required for testing FNN?
- 4) More references should be included regarding the experimental implementation of networks: Nat. Comm. 7, 13251 (2016), PRX Quantum 2, 020346 (2021), PRX Quantum 3, 030342 (2022), npj quantum information 5, 89 (2019).

Reviewer #2 (Remarks to the Author):

In the manuscript entitled "Certification of non-classicality in all links of a photonic star network without assuming quantum mechanics" the authors experimentally demonstrate the effect known as full network nonlocality. In a standard network nonlocality, it is enough that only one part of a network uses nonclassical resources, but the authors aim to certify that every link of the network exposes correlations that cannot be simulated classically. I think that the paper is well written and the topic they explore is very relevant at this moment. Quantum networks are receiving more attention and on one side it is very nice that there is an experimental demonstration involving entangling measurement on three particles and on the other side a simple enough theoretical method that can prove that the network manifests the full network nonlocality.

The only question that could have been relevant is a discussion about whether this quantum realization can be used to demonstrate also the genuine network nonlocality, another type of network nonlocality that is different than the full network nonlocality (It is introduced in <https://journals.aps.org/pr/abstract/10.1103/PhysRevA.105.022206>).

In general, there is not really much I would add to this submission and I congratulate the authors on this nice work and concise but clear presentation.

Reviewer #3 (Remarks to the Author):

In this work the authors introduce a new criteria for determining that all sources in a network must be non-classical. This criteria was then satisfied in a proof-of-concept experiment, with some additional assumptions on the lasers. Previously non-classicality was demonstrated in networks, but it is possible to demonstrate this even if only one of the sources is non-classical, and not all sources. This work describes a more demanding notion of non-classicality.

This work merits publication in Nature Communications as it highlights a new notion of non-classicality in a network. It uses the inflation technique in ways that could be useful for future study. The experiment is also non-trivial.

One possible complaint one can have is that there is a global laser source that results in photons going to three detectors. Indeed, the authors point this out and try to mitigate it when they say "to meet [the requirement of independent sources] in the experiment, we split a single laser beam into three and pump three SPDC sources in parallel." There is some additional randomisation of the local beams. However, one could ultimately say this is not in the spirit of network non-classicality. Therefore it is not a true experimental demonstration of the theory. That being said, it should inspire further experiments that have independent light sources, which would be extremely challenging to synchronise signals in the entangled measurements.

We are also grateful to the reviewers for their careful reading of our paper and their helpful feedback. We are pleased to see that they acknowledge that ours is a “challenging and interesting experiment”, that “the topic they explore is very relevant at this moment”, and that their opinion is that “this work certainly deserves publication in a broad-impact journal”. Below we address the minor points raised by the reviewers, writing also the revisions we have made to our manuscript in order to do so.

Reply to Reviewer 1

We thank Reviewer 1 for their time spent in reviewing our manuscript and for the concrete suggestions of changes.

Comment 1: *“The authors should comment on the postselection performed when the Bell state measurement is carried out. In particular about the device-dependent nature of the experiment.”*

Response: We thank Reviewer 1 for this important comment. We would like to briefly note that the inequalities obtained, despite only depending on one of the central party’s outcomes, constitute a device-independent witness of FNN: regardless of which particular states the parties share and which particular measurements they perform, if they observe a violation of \mathcal{I}_1 , \mathcal{I}_2 and \mathcal{I}_3 they can guarantee the presence of FNN. This being said, Reviewer 1 is right in that, in our concrete experimental realisation, we need postselection in order to carry out the measurement in the central node. In fact, only by using only linear optics as we do, one always needs postselection in order to realise entangled measurements. This opens the postselection loophole (see [DOI:10.1103/PhysRevLett.83.2872]) in most current multiphoton nonlocal experiments. In the new version of our manuscript we are adding a discussion, in the section regarding experimental results, to clarify that our experimental demonstration is open to the postselection loophole, in addition to the locality loophole.

Comment 2: *“It would be useful for the reader to have in Fig. 3b the theoretical values expected for the inequalities exploited.”*

Response: We thank Reviewer 1 for this suggestion, with which we agree. For this reason, in the new version of the manuscript we are adding the description of the theoretical maximum quantum values achievable in the caption of Fig. 3, in addition to the arrow in Fig. 3b that was in the initial version of the figure.

Comment 3: *“It is not clear to me which fidelity is required to have a simultaneous violation of the inequalities. In particular, on page 10 it is discussed a visibility $v=0.88$ and $v=0.89$ while the tomography reveals a visibility of the GHZ state of $v=0.82$. What is the connection between the experimental visibility and the one required for testing FNN?”*

Response: We are thankful to Reviewer 1 for raising this point, since it allows us to improve the clarity of our manuscript. On one hand, the visibility $v = 0.88$ corresponds to the amount of white noise that each of the EPR pairs generated in the sources can tolerate and still observe FNN, assuming an ideal GHZ measurement. We report visibilities above 97%, thus satisfying this requirement. On the other hand, when considering a noise model with ideal sources but white noise in the GHZ measurement performed by the central party, the required fidelity for the GHZ measurement in order to detect FNN with the inequalities derived is calculated to be 0.763. In this case, we report an experimental fidelity above 82%. In the new version of our manuscript we are addressing these matters more explicitly in the section regarding experimental details, in order to avoid potential confusions.

Comment 4: *“More references should be included regarding the experimental implementation of networks: Nat. Comm. 7, 13251 (2016), PRX Quantum 2, 020346 (2021), PRX Quantum 3, 030342 (2022), npj quantum information 5, 89 (2019).”*

Response: We thank Reviewer 1 for these pointers to relevant literature. In the new version of our manuscript we are adding these references as Refs. [25, 29, 32, 33].

Reply to Reviewer 2

We are grateful to Reviewer 2 for their time invested in reviewing our manuscript and for their acknowledgment of our contributions both in the theory and experimental fronts.

Comment: *“The only question that could have been relevant is a discussion about whether this quantum realization can be used to demonstrate also the genuine network nonlocality, another type of network nonlocality that is different than the full network nonlocality (It is introduced in [DOI:10.1103/PhysRevA.105.022206]).”*

Response: We thank Reviewer 2 for this comment. It raises an interesting question since, as is discussed in Section VI.A of [DOI:10.1088/1361-6633/ac41bb], full network nonlocality (FNN) and genuine network nonlocality (GNN) are distinct forms of network nonlocality, in that there exist FNN correlations that are not GNN, and GNN correlations that are not FNN. Unfortunately, GNN is notably more difficult to characterise, analytically and numerically, than FNN, and thus our decision to focus on FNN in the first place.

Reply to Reviewer 3

We thank Reviewer 3 for their time spent in reviewing our manuscript and for their explicit recommendation for publication.

Comment: *“One possible complaint one can have is that there is a global laser source that results in photons going to three detectors. Indeed, the authors point this out and try to mitigate it when they say ”to meet [the requirement of independent sources] in the experiment, we split a single laser beam into three and pump three SPDC sources in parallel.” There is some additional randomisation of the local beams. However, one could ultimately say this is not in the spirit of network non-classicality. Therefore it is not a true experimental demonstration of the theory. That being said, it should inspire further experiments that have independent light sources, which would be extremely challenging to synchronise signals in the entangled measurements.”*

Response: Reviewer 3 is right in that, in our experiment, we use the same pump laser for feeding the three sources of EPR states. Indeed, the motivation for this is precisely that acknowledged by Reviewer 3, namely the extreme difficulty of synchronising signals coming from three truly independent sources. In the new version of the manuscript we have briefly extended our discussion on this matter. Despite this, we want to note that, in addition to randomising the phases of the three beams split from the pump laser with independent quantum random number generators, we report in the Supplementary Note 2 how the experimental independence of the sources manifests itself in the factorisation of marginals of the empirical distribution, finding for all such marginals values compatible with the hypothesis of truly independent sources.